# Effect of Ionizing Radiation from Computed Tomography on Differentiation of Human Embryonic Stem Cells into Neural Precursors

**DOI:** 10.3390/ijms20163900

**Published:** 2019-08-10

**Authors:** Christine Hanu, Burk W. Loeliger, Irina V. Panyutin, Roberto Maass-Moreno, Paul Wakim, William F. Pritchard, Ronald D. Neumann, Igor G. Panyutin

**Affiliations:** 1Department of Radiology and Imaging Sciences, Clinical Center, NIH, 10 Center Dr., Bethesda, MD 20892, USA; 2Biostatistics and Clinical Epidemiology Service, Clinical Center, NIH, 10 Center Dr., Bethesda, MD 20892, USA

**Keywords:** human embryonic stem cell differentiation, low-dose radiation, computed tomography, PAX6, nestin

## Abstract

We studied the effect of radiation from computed tomography (CT) scans on differentiation of human embryonic stem cells (hESCs) into neuronal lineage. hESCs were divided into three radiation exposure groups: 0-dose, low-dose, or high-dose exposure. Low dose was accomplished with a single 15 mGy CT dose index (CTDI) CT scan that approximated the dose for abdominal/pelvic CT examinations in adults while the high dose was achieved with several consecutive CT scans yielding a cumulative dose of 500 mGy CTDI. The neural induction was characterized by immunocytochemistry. Quantitative polymerase chain reaction (qPCR) and Western blots were used to measure expression of the neuronal markers *PAX6* and *NES* and pluripotency marker *OCT4*. We did not find any visible morphological differences between neural precursors from irradiated and non-irradiated cells. However, quantitative analyses of neuronal markers showed that *PAX6* expression was reduced following exposure to the high dose compared to 0-dose controls, while no such decrease in *PAX6* expression was observed following exposure to the low dose. Similarly, a statistically significant reduction in expression of *NES* was observed following high-dose exposure, while after low-dose exposure, a modest but statistically significant reduction in *NES* expression was only observed on Day 8 of differentiation. Further studies are warranted to elucidate how lower or delayed expression of *PAX6* and *NES* can impact human fetal brain development.

## 1. Introduction

The first few weeks of human development are considered to be the most sensitive and vulnerable to a number of teratogenic agents, including ionizing radiation [1,2]. Therefore, exposure of women to ionizing radiation—as may occur during CT scans—while in the early stages of pregnancy is a subject of concern [3]. Some of the major differentiation events that contribute to human embryonic brain development begin around the third week of gestation [2], a time when pregnancy is not yet confirmed in most women. The International Commission on Radiological Protection has reported embryo and fetal brain malformations, such as microcephaly and growth retardation at doses above 100 mGy [4]. However, there is evidence of radiation effects on embryogenesis at doses of 100 mGy or lower from animal studies [5,6] using cell models [7], in humans exposed to diagnostic test radiation [8], as well as in A-bomb survivors exposed prenatally [9,10,11,12].

Pluripotent human embryonic stem cells (hESCs) could be a useful in vitro model which can serve as a surrogate for the preimplantation embryo in assessing the biological effects of ionizing radiation during early brain development. A previous study showed that exposure to 10–30 mGy of γ-radiation attenuated differentiation of immature mouse-derived neural stem cells to glial cells [13]. Other work using human neural progenitor cells (NPCs) has shown altered neural development and gene expression following chronic γ-ray exposure [7]. These authors reported that 496 mGy of γ-irradiation delivered over a duration of 72 h can affect neuronal differentiation, whereas altered gene expression was observed at a dose lower than 100 mGy. Another group using in vitro models studied the responsiveness of hESCs, derived neural stem cells (NSCs), and mature neurons to high-dose ionizing radiation [14]. They found that mature neurons were the most resistant, followed by the NSCs, and ending with the hESCs as the least resistant to radiation. Radiation effects on prenatal development have also been shown to affect differentiation of hESCs into other lineages such as cardiomyocytes [15] and definitive endoderm [16]. To date, there are no other studies exploring radiation responses from CT exposure in hESCs directed to differentiate towards NPCs.

In this study, we used an efficient and reliable neural differentiation technique to study the effect of ionizing radiation exposure on the differentiation of hESCs into neural progenitors. The morphology of the neural rosettes, NPCs, and neural progenitors was assessed along with the expression of the neural development markers paired box protein 6 (*PAX6*) and nestin (*NES*) [17]. Based upon previous in vitro studies [18,19], we hypothesized that levels of the pluripotency marker octamer-binding transcription factor 4 (*OCT4*) would initially be high at the start of neural induction with subsequently lower levels observed thereafter. Expression levels of the neural differentiation markers, *PAX6* and *NES*, would be low at the start of differentiation and increase as NPCs were established in irradiated and non-irradiated cultures. Our results will hopefully contribute to the limited data available, and further our understanding of radiation effects on neural differentiation from imaging procedures.

## 2. Results

### 2.1. Neural Morphologies Appear to Develop Normally in Irradiated Samples

Colonies of H9 hESC were exposed to a 0-dose (control), low dose (15 mGy CTDI), or high dose (500 mGy CTDI) of CT radiation, and differentiated into neuronal lineage. The doses measured by ionization chamber were 0, 17, and 572 mGy, respectively. The timeline of the differentiation protocol is shown in Figure 1A. Twenty-four hours after exposure to radiation, colonies of hESCs were dissociated into single cell suspensions (Figure 1B, Day 0), and seeded into AggreWell plates with uniformly-sized wells that hold a single embryoid body (EB) (Figure 1B, Day 1).

On Day 5, EBs were transferred to Matrigel-coated 6-well plates. These EBs were already stained positive for a neural precursor marker, PAX6, and a neural differentiation marker, nestin, in both non-irradiated and irradiated samples (Figure 2). On Day 7, specific structures, so-called neural rosettes with intense nestin staining in their centers, appeared on the surfaces of the EBs (Figure 1B and Figure 2). By Day 15, the neural rosettes mostly dissociated into cells in monolayer culture with typical NPCs morphology that stained positive for both PAX6 (nuclei) and nestin (cytoplasm) (Figure 2). Afterwards, the media was changed to STEMdiff Neural Progenitor Medium, and cells were maintained for another week while they developed characteristic neuronal features positive for PAX6 and neuronal marker β-tubulin III (Figure 2, Day 23). We could not detect any morphological differences in the cultures obtained from 0-dose controls and those exposed to low or high dose. Therefore, we concluded that even after exposure to high-dose irradiation, hESCs were still able to differentiate into neural precursors.

### 2.2. Effect of CT Irradiation on Expression of Neuronal Markers

We then quantitatively assessed expression of neural differentiation markers in three independent rounds of differentiation, i.e., Trials 1, 2, and 3. Panels A–C in Figure 3, Figure 4 and Figure 5 show the timeline of relative RNA expression of neuronal markers *PAX6*, *NES*, and a pluripotency marker *OCT4*, for all three trials, respectively. The results of the statistical analyses for *PAX6*, *NES*, and *OCT4* are shown in Figure 3D, Figure 4D, and Figure 5D and Table 1, Table 2, Table 3, Table 4, Table 5 and Table 6.

Both non-irradiated and irradiated cells showed an increase in *PAX6* expression throughout differentiation (Figure 3). In the low-dose groups, *PAX6* exhibited RNA expression levels comparable to levels in the 0-dose groups. By contrast, the high-dose groups showed substantially decreased *PAX6* expression over the majority of the experimental time course. In Trials 1 and 3, *PAX6* RNA expression levels in high-dose groups became almost equal to the levels in the 0-dose and low-dose groups by Day 12, while in Trial 2 *PAX6* RNA expression was substantially reduced across the entire time course. Statistical analysis confirmed that there is no difference between 0-dose and low-dose groups in *PAX6* RNA expression (Table 1). However, *PAX6* RNA expression was significantly lower in the high-dose groups on Days 4, 6, 8, 10, and 12 than in the 0-dose controls (Table 2).

Expression of *NES* was somewhat variable RNA between the trials. The expression level of *NES* RNA increased after day 2 or 4 reaching a maximum on day 6 in Trial 1, and on day 8 in Trials 2 and 3. *NES* RNA expression then decreased especially in Trials 1 and 2. Overall *NES* RNA expression was reduced in the high-dose groups compared to 0-dose controls (Figure 4). Statistical analysis showed that *NES* RNA expression was similar in low-dose and 0-dose groups on Days 0, 2, 6, 10, and 12; however, the difference in RNA expression between the groups was statistically significant on Day 8 and borderline significant on Day 4 (Table 3). *NES* RNA expression was similar in high-dose and 0-dose groups on Days 0 and 2, and then reduced in high-dose groups starting from Day 4 (Table 4).

*OCT4* RNA expression showed the expected decrease until Day 4, when expression had become virtually undetectable (Figure 5). Statistical analysis showed that exposure to CT irradiation did not have any statistically significant effect on *OCT4* expression levels (Table 5; Table 6).

Evidence of reduced *PAX6* expression was supported by Western blot analysis, which showed reduced PAX6 protein expression in high-dose relative to 0-dose groups (Figure 6). Analysis of band intensities indicated a decrease of PAX6 expression of approximately 50%, which agrees with qPCR data.

## 3. Discussion

There are conflicting findings resulting from research investigating the brain carcinogenic risk from low dose CT scans. One study demonstrated that a cumulative dose of 60 mGy may triple the risk of brain cancer in children [20] while another study suggested that CT-related brain cancer risk was confounded by insufficient controls for cancer susceptibility syndromes [21]. Retrospective studies mostly based on A-bomb survivors suggested that the processes involved in neural development could be especially sensitive to radiation exposure [1,9,10,11]. More recent studies of women that had undergone CT scans in the early stages of pregnancy suggested that exposure of a blastocyst could result in “all or none” phenomena, i.e., either miscarriage or no consequences [22,23]. However, very little experimental data exists regarding the effects of the low radiation doses used in common radiological imaging techniques on neurological development. Greater understanding in this regard has the potential to have major implications for the official health recommendations of diagnostic protocols, especially in the case of women who may be pregnant at the time of such scans.

This study aimed to better discern the effects of low-level radiation on neural differentiation by inducing neural cell fates in hESCs after exposure to different doses of radiation. We found that hESCs exposed to CT scans with an absorbed dose of 17 mGy and even to 572 mGy were still able to differentiate into NPCs forming characteristic structures, neural rosettes, and expressing neurospecific markers such as PAX6 and NES. They also were able to differentiate further into neural precursors exhibiting characteristic neuronal morphology and expressing the neuron-specific marker β-tubulin III. However, the relative expression level of *PAX6* was reduced almost 2 times in cells exposed to 572 mGy compared to 0-dose controls; there was no reduction in relative expression after exposure to 17 mGy. Similar results were observed for *NES*; the relative expression level of this NPC marker was reduced after exposure to 572 mGy. The reduction was not as pronounced as for the *PAX6* but it was statistically significant. After exposure to 17 mGy, a modest but statistically significant reduction in the *NES* relative expression level was observed only on Day 8.

We can only speculate if the exposure of human blastocysts to a 572 mGy dose would result in a similar reduction of *PAX6* expression and how this would affect embryo development. Permanent reduction in *PAX6* expression levels is known to cause aniridia in eyes of *PAX6^+/−^* patients [24]. Less is known about temporal downregulation of PAX6. Interesting results were obtained recently in a mouse model that allowed for inducible Pax6 depletion [25]. However, caution should be taken when extending results obtained in mice experiments to human patients because it was found that PAX6 expression patterns are quite different during neuroectoderm development between mice and humans in early stages of embryogenesis [17]. An interesting clue as to a possible reason of *PAX6* downregulation after CT irradiation comes from recent experiments in zebrafish [26]. The authors of this study found that at low dose-rate exposure to γ-irradiation the most affected signaling pathway was retinoic acid receptor activation. Retinoic acid is a key regulator of PAX6 expression [27,28]. Further experiments in hESCs may shed light on whether the retinoic acid pathway is involved in the downregulation of PAX6 after CT irradiation exposure of human cells.

We could not find evidence in the literature of PAX6 involvement in regulation of *NES* expression. Therefore, downregulation of *NES* after exposure to 572 mGy most likely is not a result of PAX6 downregulation. The reduction in expression of both genes may have a common underlying mechanism. Future experiments using whole genome gene expression analysis could reveal more information about cellular pathways that are affected by exposure to CT irradiation.

## 4. Materials and Methods

### 4.1. Cell Line & Cell Culture

All methods were carried out in accordance with relevant NIH guidelines and regulations. H9 hESCs (WA09 cell line, passage 21) were purchased from WiCell Research Institute (Madison, WI, USA). This cell line is registered in the NIH Human Embryonic Stem Cells Line Registry (https://grants.nih.gov/stem_cells/registry/current.htm). All experiments with this cell line were performed according to NIH Guidelines for Human Stem Cell Research. H9 hESCs were cultured in feeder-free complete mTeSR-1 culture medium (STEMCELL Technologies, B.C., Canada) and routinely passaged every 5 to 7 days with collagenase IV (STEMCELL Technologies) onto Matrigel (Corning, NY, USA)-coated flasks. H9 hESCs were seeded onto T-75 Matrigel-coated flasks 48 h prior to irradiation (Figure 1A). The cells were either 0-dose controls or were exposed to radiation before initiating neural induction.

### 4.2. CT Scan Irradiation

Cell irradiations were performed using a single X-ray tube of a dual-source CT scanner (SOMATOM Definition Flash, Siemens Healthcare, Forchheim, Germany). All flasks were brought to the scanner just prior to irradiation and were returned to the CO_2_ incubator immediately following irradiations. Flasks were placed into a 3D torso phantom (model 602, CIRS, Norfolk, VA, USA) and placed at the scanner isocenter (Appendix A). Cells in the low-dose group were irradiated with a dose of 15 mGy CT dose index (CTDI), approximating the dose for abdominal/pelvic CT examinations in adults, delivered as a single axial scan at 120 kVp and a tube current of 200 mAs. As positive controls, cells in the high-dose group were irradiated at a dose of 500 mGy CTDI, delivered as 10 consecutive axial scans at 120 kVp and a tube current of 666 mAs.

The CTDI values do not generally reflect the true absorbed dose to the samples placed in the gantry. The absorbed dose inside the phantom where the cells were to be placed was measured using a calibrated 10 cm-long ionization chamber (Fluke, Victoreen 660 with 660-6 probe Columbus, OH, USA). The doses measured by the probe were calculated as 17 and 572 mGy for low- and high-dose exposures, respectively.

Viability of hESCs after exposure to radiation was similar to that reported previously [29]. Herein, the reaction of hESCs to CT irradiation was assessed by the expression of pluripotency marker OCT4 and the DNA repair foci marker 53BP1 [19]. Appendix A shows that hESC colonies stained positive with OCT4 antibody, i.e., cells did not lose pluripotency immediately after exposure. Staining hESCs colonies with 53BP1 antibody one hour after irradiation showed that cells responded to CT exposure with formation of DNA repair foci in the dose dependent manner [19]. In a separate experiment we exposed neural precursor cells on Day 25 of differentiation to 0, low, or high dose. These differentiated cells also responded to CT exposure by formation of DNA repair foci revealed with 53BP1 antibody staining one hour after irradiation (Appendix A).

### 4.3. Generation of Neural Progenitor Cells from hESCs

Studies were performed using three independent biological replicates, Trials 1, 2, and 3, taking care to reproduce conditions and procedures for all the steps of the study. The cells were allowed to recover for 24 h after irradiations. Then, we used the STEMdiff Neural Induction Protocol and reagents by STEMCELL Technologies (Figure 1A). Briefly, the cells were dissociated into a single cell suspension, and evenly distributed onto AggreWell^TM^ 800 plates in 2 mL of neural induction medium (NIM) supplemented with 10 µM of Y-27632 ROCK inhibitor. Partial medium changes were performed from Day 1 to 4 with NIM culture medium. Embryoid bodies (EBs) were harvested on Day 5 of differentiation and plated onto Matrigel-coated 6-well plates. Full NIM culture medium changes were performed daily from Day 6 to 11 of differentiation.

By Day 7 of differentiation, the EBs transformed into neural rosettes that were isolated and seeded onto a Matrigel-coated 6-well plate. By Day 17 to 19 of differentiation, the neural precursor cells had grown between the neural rosette clusters to form a monolayer. They were harvested at 80–90% confluency and plated onto Matrigel-coated 6-well plates in NIM culture medium. The cells were allowed to grow for an additional 4 to 5 days in neural induction medium. Then, the cells were grown in supplemented STEMdiff neural progenitor basal medium (STEMCELL Technologies).

### 4.4. Immunocytochemistry

Day 5 EBs were plated onto Matrigel-coated chambered microscope slides (Falcon, VWR, USA), and were then subjected to the differentiation procedure described above. At the indicated timepoints, the cells on the slides were fixed with 4% paraformaldehyde and permeabilized with 0.1% Triton X-100. The samples were blocked with 5% bovine serum albumin and stained with primary antibodies overnight at 4 °C followed by secondary antibodies and incubated at room temperature for 60 min. EBs were stained with the following antibodies: OCT4 (Stem Cell Technologies, 1:250), nestin (Stem Cell Technologies, 1:1000), and PAX6 (Invitrogen, Waltham, MA, USA, PA1-801, 1:50). Neural rosettes were stained with PAX6 and nestin antibodies. Neural progenitors were stained with β**-**tubulin III  antibodies (Stem Cell Technologies, 1:125) and PAX6 antibodies. We also used 53BP1 Polyclonal Antibody (PA1-16565, Thermo Fisher Scientific). Primary antibodies were followed with their appropriate secondary antibodies Alexa Fluor^®^ 488 Goat anti-mouse IgG (Invitrogen, A-11011, 1:100) and Alexa Fluor^®^ 594 Goat anti-rabbit IgG (Invitrogen, A-11012, 1:100). All samples were imaged with the Zeiss Axiovert 200 fluorescence microscope.

### 4.5. Quantitative Real-Time PCR

H9 hESCs and NPCs were harvested with Accutase. EBs and neural rosettes were removed with a cell scraper and manually broken down to a single cell suspension using a pestle followed by passing the cells through a 21-gauge needle. All cell pellets were stored in PBS and RNA*later* solution (Invitrogen) for RNA extraction. Total RNA was extracted and purified using the PureLink RNA Mini Kit (Invitrogen), and DNA-*free*™ DNA Removal Kit (Invitrogen) to remove genomic DNA. RNA quality was assessed on the Agilent RNA 2100 Bioanalyzer (Agilent, Santa Clara, CA, USA); RNA integrity number of the samples was more than 8.0. TaqMan^®^ RNA-to-C_T_™ 1-step kit (Applied Biosystems, Waltham, MA, USA) was used to assess gene expression. StepOne™ Plus Real-Time PCR System in 96-well plate format was used to measure gene expression (Applied Biosystems). The following TaqMan primers were used: *OCT4* (Applied Biosystems, Hs04260367_gH), *PAX6* (Applied Biosystems, Hs01088112_m1), and *NES* (Applied Biosystems, Hs04187831_g1). For relative quantification (RQ), the gene expression levels of *OCT4*, *PAX6*, and *NES* were normalized against the TATA-binding protein (TBP) housekeeping gene (Applied Biosystems, Hs00427620_m1).

### 4.6. Western Blots

Protein was extracted from cells by using RIPA Buffer (Thermo Scientific, Waltham, MA, USA), and subsequently quantified with the Pierce™ BCA Protein Assay Kit (Thermo Scientific). Twenty micrograms of total proteins per lane were separated on a 4–12% Bis-Tris gel, and transferred onto a polyvinylidene difluoride (PVDF) membrane (Thermo Scientific). The membrane was blocked and incubated with primary antibodies overnight at 4 °C (PAX6 monoclonal antibody: Invitrogen, MA1-109, 1:100; anti-OCT4, Abcam, Cambridge, MA, USA, 1:100) followed by secondary antibodies (goat anti-mouse Alexa Fluor Plus 488: Invitrogen, 1:1000; goat anti-rabbit Qdot 605, Invitrogen, 1:50) for 60 min at room temperature. All blots used β-actin (Abcam, 1:500) as a positive control. Three independent Western blot experiments were performed. Blots were imaged using a Fujifilm Fluoro Image Analyzer FLA-5000. Protein band densities were quantified using Fujifilm Science Lab Image Gauge Ver. 4.0.

### 4.7. Statistical Analysis

The statistical analysis addressed the question of whether relative RNA expression (RQ) of *PAX6*, *NES*, and *OCT4* are different for the three radiation doses over time. Analyses were done separately for RNA expressions of *PAX6*, *NES*, and *OCT4*, with each analysis fitting all relevant data into one statistical mixed model.

RNA expression levels between trials, as well as between doses, were assumed to be independent. Replicates within each day were assumed to be correlated, and so were RQs over time within each dose and within each trial. As a result, a mixed model that allows for correlations between measurements was used. The response (dependent) variable was the RQ; and the explanatory (independent) variables were Trial (1, 2 or 3), Dose (0, low or high), Day (0–12), and Dose-by-Day interaction. The categorical Day effect and Dose-by-Day interaction allowed for individually fitted RQs for each timepoint (day) and each dose, without any assumption regarding the shape of the time trend. To show a full picture of RNA expression levels over time, per dose, from all trials and replicates combined, RQ least square means calculated from the model were plotted with 95% confidence intervals. Differences in RQ least square means were tabulated for each day. The SAS statistical software (SAS Institute, Cary, NC, USA) was used for this analysis.

The datasets generated during and/or analyzed during the current study are available from the corresponding author on reasonable request.

## Figures and Tables

**Figure 1 ijms-20-03900-f001:**
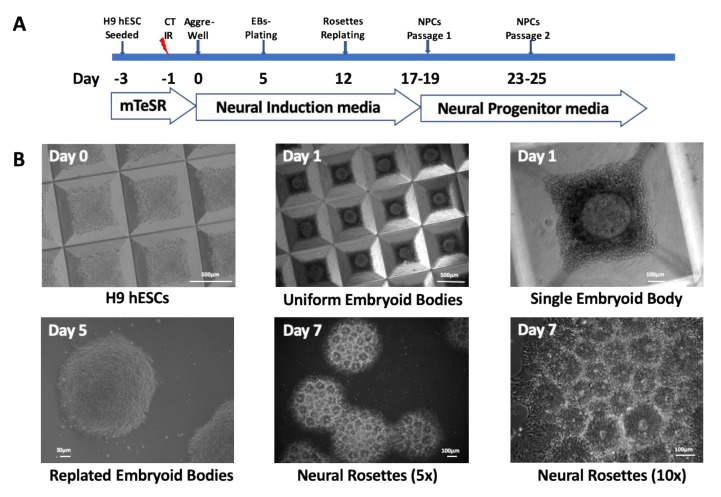
(**A**) Timeline of radiation exposure and subsequent cell culture passaging. (**B**) Development of neural morphologies as human embryonic stem cells (hESCs) progress to embryoid bodies and then neural rosettes.

**Figure 2 ijms-20-03900-f002:**
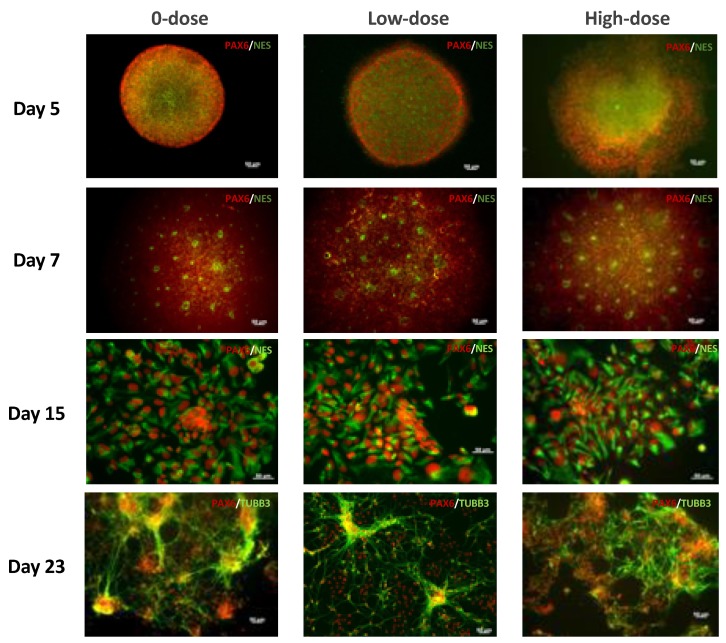
Images of hESCs at different stages of differentiation to neuronal lineage. Staining of embryoid bodies (EBs) (Day 5); neural rosettes (Day 7); human neural progenitor cells (NPCs) (Day 15); and neural precursors (Day 23) with neuronal markers as indicated on the panels. Scale bar = 50 µm.

**Figure 3 ijms-20-03900-f003:**
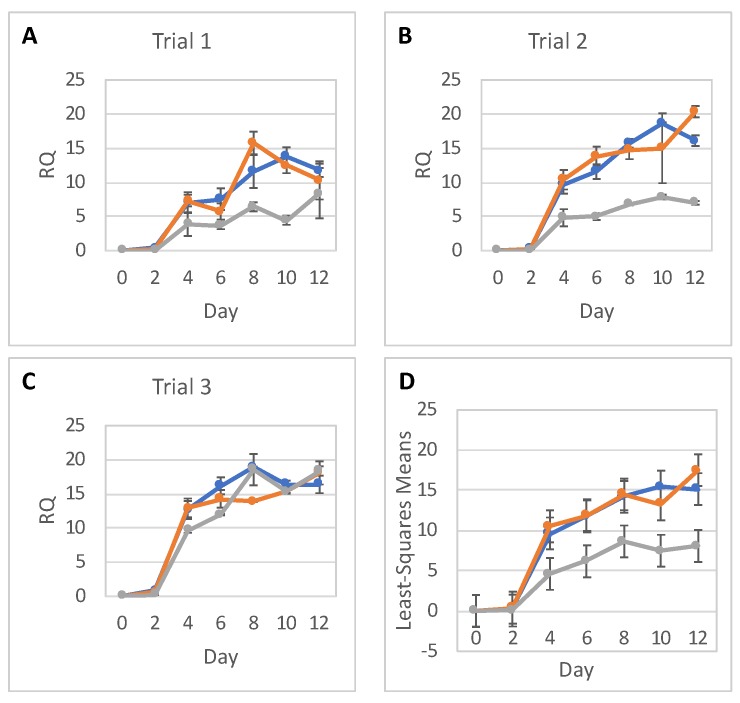
*PAX6* gene expression. *PAX6* relative to *TBP* RNA expression (RQ) in hESCs during their differentiation into NPCs in three independent trials (**A**–**C**) in 0-dose (blue), low-dose (orange), and high-dose (grey) samples; error bars show standard deviations of experimental measurements. (**D**) Statistical analysis of *PAX6* expression data; error bars show 95% confidence limits for the least square means of *PAX6* RQs in 0-dose (blue), low-dose (orange), and high-dose (grey) samples from all trials and with replicates combined.

**Figure 4 ijms-20-03900-f004:**
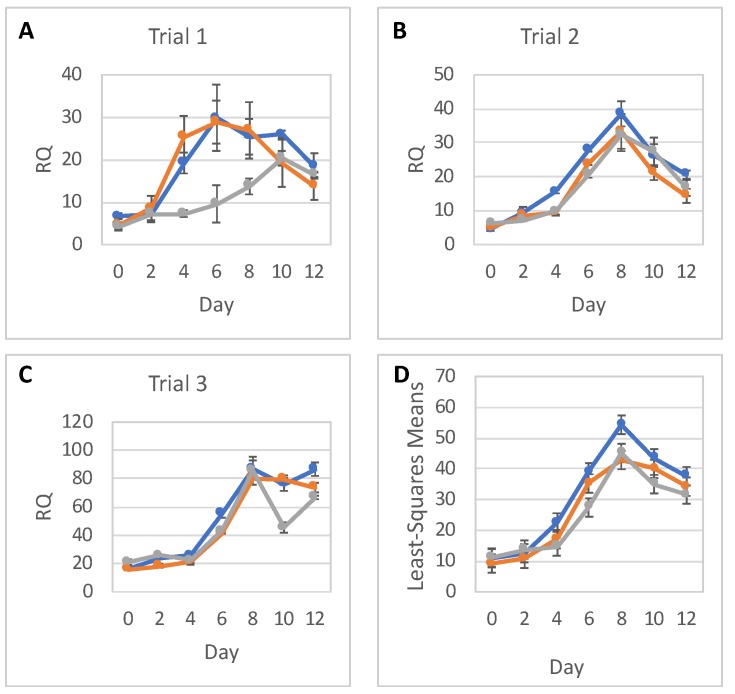
*NES* gene expression. *NES* relative to *TBP* RNA expression (RQ) in hESCs during their differentiation into NPCs in three independent trials (**A**–**C**) in 0-dose (blue), low-dose (orange), and high-dose (grey) samples; error bars show standard deviations of experimental measurements. (**D**) Statistical analysis of *NES* expression data; error bars show 95% confidence limits for the least square means of *NES* RQs in 0-dose (blue), low-dose (orange), and high-dose (grey) samples from all trials and with replicates combined.

**Figure 5 ijms-20-03900-f005:**
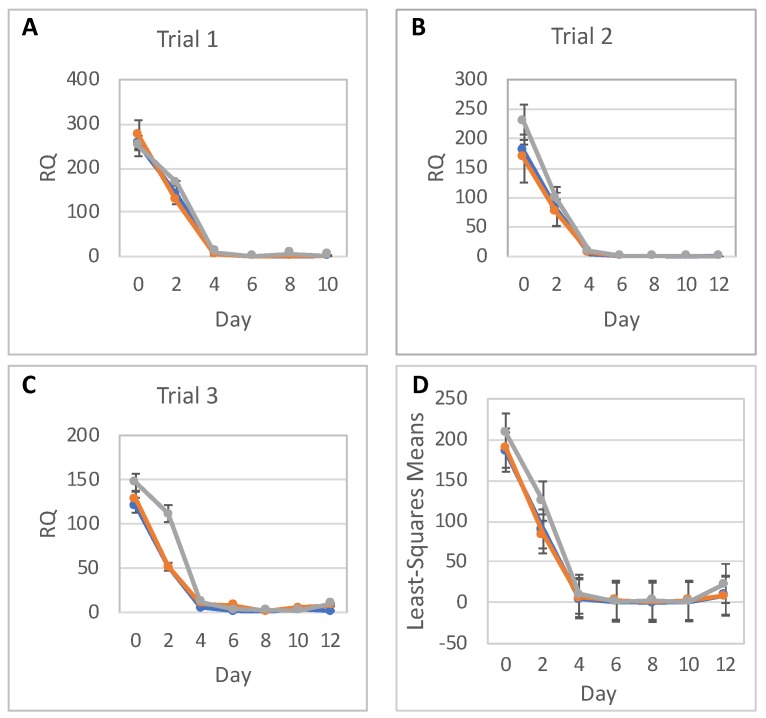
*OCT4* gene expression. *OCT4* relative to *TBP* RNA expression (RQ) in hESCs during their differentiation into NPCs in three independent trials (**A**–**C**) in 0-dose (blue), low-dose (orange), and high-dose (grey) samples; error bars show standard deviations of experimental measurements. (**D**) Statistical analysis of *OCT4* expression data; error bars show 95% confidence limits for the least square means of *OCT4* RQs in 0-dose (blue), low-dose (orange), and high-dose (grey) samples from all trials and with replicates combined.

**Figure 6 ijms-20-03900-f006:**
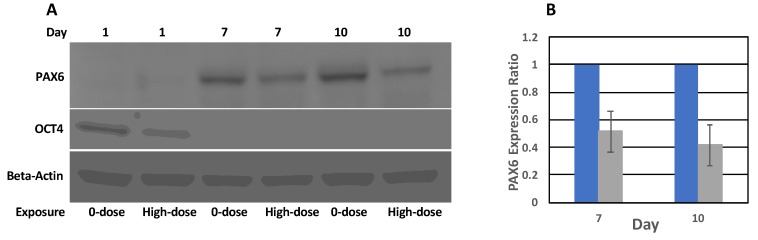
PAX6 and OCT4 protein expression. (**A**) Western blot analysis of PAX6, OCT4, and β-actin protein expression in 0-dose and high-dose samples. Images of the same membrane probed by different antibodies are presented. (**B**) Quantitation of Western blot data of PAX6 expression on Days 7 and 10 in high-dose (grey) samples relative to 0-dose (blue) samples. Error bars show standard deviations of three Western blot experiments.

**Table 1 ijms-20-03900-t001:** Difference between 0-dose and low-dose groups in least square means of *PAX6* RNA expression according to day.

Day	Estimate	Lower CL	Upper CL	*p*-Value
0	0.0036	−2.79	2.80	0.998
2	0.0018	−2.79	2.80	0.999
4	−0.8870	−3.68	1.91	0.529
6	−0.1509	−2.95	2.65	0.915
8	−0.2970	−3.09	2.50	0.833
10	2.2233	−0.57	5.02	0.117
12	−2.3540	−5.15	0.44	0.098

CL = confidence limit for the difference.

**Table 2 ijms-20-03900-t002:** Difference between 0-dose and high-dose groups in least square means of *PAX6* RNA expression according to day.

Day	Estimate	Lower CL	Upper CL	*p*-Value
0	0.0109	−2.79	2.81	0.994
2	0.3522	−2.44	3.15	0.802
4	5.0219	2.23	7.82	0.001
6	5.5679	2.77	8.36	0.0002
8	5.5338	2.74	8.33	0.0002
10	7.9943	5.20	10.79	<0.0001
12	7.0801	4.28	9.88	<0.0001

CL = confidence limit for the difference.

**Table 3 ijms-20-03900-t003:** Difference between 0-dose and low-dose groups in least square means of *NES* RNA expression according to day.

Day	Estimate	Lower CL	Upper CL	*p*-Value
0	1.7230	−2.55	6.00	0.417
2	1.7889	−2.48	6.06	0.400
4	5.4113	1.14	9.68	0.015
6	3.6849	−0.59	7.96	0.089
8	11.4302	7.16	15.70	<0.0001
10	3.4629	−0.81	7.74	0.108
12	3.4354	−0.84	7.71	0.111

CL = confidence limit for the difference.

**Table 4 ijms-20-03900-t004:** Difference between 0-dose and high-dose groups in least square means of *NES* RNA expression according to day.

Day	Estimate	Lower CL	Upper CL	*p*-Value
0	−0.2370	−4.51	4.04	0.911
2	−1.1642	−5.44	3.11	0.582
4	7.8211	3.55	12.09	0.001
6	11.4532	7.18	15.73	<0.0001
8	9.1564	4.88	13.43	0.0001
10	8.4105	4.14	12.68	0.0004
12	5.9798	1.71	10.25	0.008

CL = confidence limit for the difference.

**Table 5 ijms-20-03900-t005:** Difference between 0-dose and low-dose groups in least square means of *OCT4* RNA expression according to day.

Day	Estimate	Lower CL	Upper CL	*p*-Value
0	−4.7243	−39.99	30.54	0.786
2	6.1923	−29.07	41.46	0.722
4	−1.7173	−36.98	33.55	0.921
6	−2.3424	−37.61	32.92	0.893
8	−0.6153	−35.88	34.65	0.972
10	−1.5768	−36.84	33.69	0.928
12	1.1619	−38.89	41.22	0.953

CL = confidence limit for the difference.

**Table 6 ijms-20-03900-t006:** Difference between 0-dose and high-dose groups in least square means of *OCT4* RNA expression according to day.

Day	Estimate	Lower CL	Upper CL	*p*-Value
0	−23.3427	−58.61	11.92	0.186
2	−34.2150	−69.48	1.05	0.057
4	−5.7133	−40.98	29.55	0.743
6	−0.5544	−35.82	34.71	0.975
8	−2.9063	−38.17	32.36	0.867
10	−0.0849	−35.35	35.18	0.996
12	−14.2786	−54.33	25.78	0.473

CL = confidence limit for the difference.

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
