# Peer review of "Effect of Ionizing Radiation from Computed Tomography on Differentiation of Human Embryonic Stem Cells into Neural Precursors"

_ijms, 2019, doi:10.3390/ijms20163900_

Round 1

Reviewer 1 Report

The manuscript reports an interesting set of experimental data on the effects of low-dose irradiation on differentiation of human embryonic stem cells. With the regard of presentation I have the following comments:

As the authors have analysed the in vitro effects of irradiation on differentiation of human embryonic stem cells, in the Discussion section they should clearly emphasise that their results may not be extrapolated on the in vivo effects of exposure to ionising radiation on the developing brain.

On page 3 the authors state that they ‘could not detect any morphological differences between irradiated and non-exposed cells. It would appear that such a conclusion was made on the basis of eyeballing of images of neural rosettes over the period of time of 23 days. I do not think that this is enough. Surely, this conclusion may be based on the percentage of positive cells.

Author Response

Q:As the authors have analysed the in vitro effects of irradiation on differentiation of human embryonic stem cells, in the Discussion section they should clearly emphasise that their results may not be extrapolated on the in vivo effects of exposure to ionising radiation on the developing brain.

A:We believe that the sentence in the Discussion “We can only speculate if the exposure of human blastocysts to a 572 mGy dose would result in a similar reduction of PAX6expression and how this would affect embryo development” (line 226) adequately addresses the reviewer’s concern.

Q:On page 3 the authors state that they ‘could not detect any morphological differences between irradiated and non-exposed cells. It would appear that such a conclusion was made on the basis of eyeballing of images of neural rosettes over the period of time of 23 days. I do not think that this is enough. Surely, this conclusion may be based on the percentage of positive cells.

A:We would definitely like to quantitatively support our conclusions; however, we could not find a reliable parameter that could be measured on the Figure 2 images. For example, we found that percentage of PAX6 positive cells strongly depended on how careful was the lifting the neural rosettes on day 7, and thus it varied from trial to trial.

Reviewer 2 Report

The  authors present an interesting study on the impact of ionizing radiation on  human embryonic stem cells regarding their differentiation into the neuronal  lineage. Due to clinical observations after high dose ionizing radiation and the  vast usage of ionizing radiation in diagnostic procedures there is a high demand  for examinations on sensitive periods within pregnancy. Studies on hESC might be  a model to close the gap between animal studies and real-life situations. I  agree and support the statement of the authors that for highly sensitive  molecular changes the animal experiment might be misleading and that we have to  look for models closer to the human systems.

General  remarks:

The  model is interesting, but the sensitive phase of pregnancy is later. So I would  be very careful with interpretation. I encourage the authors to cultivate longer  and to use the maturation phase after week three and to repeat the examinations  in this time frame of greater interest.

Furthermore,  I don`t understand the results regarding the repair process. Was the stain of  53BP1 performed 3 weeks after irradiation?  Usually the signal is lost within  few hours. This has to be explained. When finding this results after 3 weeks deeper examination is needed. If the stain was done close to radiation (within 1 day) the whole section of material and methods has to be checked with great caution.

The  results regarding quality controls have to be displayed in more  detail.

Special:

2.1  and 4.2 Beside no differences in morphological pattern: is there a difference in  growth rate, size of EBs,…

2.2  and 4.5.

Quality  data for RNA are missing. The whole protocol should be displayed. Furthermore  was the housekeeping gene confirmed in the system examined.

Repair process: see above.

Discussion

198  to 201: Why discussing cancer risks regarding neural development? The studies  mentioned are on childhood exposure to my knowledge – so why discussing this  topic in this manuscript?

The  authors should discuss the limits of their study – time points, single cell  experiments without interaction to other cells or the influence of the  mother.

Author Response

Q:The  model is interesting, but the sensitive phase of pregnancy is later. So I would  be very careful with interpretation. I encourage the authors to cultivate longer  and to use the maturation phase after week three and to repeat the examinations  in this time frame of greater interest.

A:We certainly would like to pursue the avenues suggested by the reviewer in our future studies. We believe that the sentence “We can only speculate…” (line 226) addresses the reviewer’s concern.

Q:Furthermore,  I don`t understand the results regarding the repair process. Was the stain of  53BP1 performed 3 weeks after irradiation?  Usually the signal is lost within  few hours. This has to be explained. When finding this results after 3 weeks deeper examination is needed. If the stain was done close to radiation (within 1 day) the whole section of material and methods has to be checked with great caution.

A:Figure 2S shows hESC colonies irradiated before differentiation; Figure 3S shows neural precursor cell irradiated on day 25 of differentiation. In both cases staining with 53BP1 antibody was done one hour after irradiation. We made corresponding changes in the Methods section (section 4.2, paragraph 3) to clarify these details.

Reviewer 3 Report

·       This manuscript aims to discuss the results of a simple question as to whether radiation exposure at CT level affects the differentiation of human embryonic stem cells to neuronal lineage. The authors attempted to demonstrate the effect of (maybe single) low dose radiation, such as CT abdominal radiography, on hESCs differentiation, but the results remain in the list of preliminary data for qPCR experiments. This manuscript seems to be a simple list of preliminary data, and has not shown in depth research results. Rather, an explanation of the association between neural cell differentiation marker expression and supplementary data such as 53BP1 discussed in the material and methods session of the manuscript may provide answers to the reader's curiosity.

·       In detail, the authors simply listed the results of each result from three individual qPCR experiments of three genes, and describe the difference between each experiment. However, there was no statistical difference in  the average of the three experiments. Also, there are no explanations on why different result occurred from 3 trials. In general, three or more experiments are carried out in every in vitro study, taking into account the experimental environmental errors that may occur in the experiment, and whether the average value has statistical significance is explained. In addition, the authors also stated that qPCR and WB analysis were performed. However, authors did not explain enough about the results of WB. Even WB shows only one test result without mean or SD. The authors should perform more experiments including western blotting and show their opinion with statistical analysis. 

·       Minor points

    -The abstract should be more specific and clearly rewritten. For example, you should equally comment on the results of qPCR and WB in order of experiment. It is also recommended that qPCR be described identically for each gene by gene / dose / time point sequence.

    - line 47-49. It is also necessary to make sure whether the "previous studies" have more than 2 references or not.

    - line 99, In the results section, the period (.) must be deleted from "2.2 Effect of CT irradiation on expression markers."

Author Response

Q:Rather, an explanation of the association between neural cell differentiation marker expression and supplementary data such as 53BP1 discussed in the material and methods session of the manuscript may provide answers to the reader's curiosity.

A:We clarify the 53BP1 staining experiments in the Methods section 4.2, last paragraph

Q: In detail, the authors simply listed the results of each result from three individual qPCR experiments of three genes, and describe the difference between each experiment. However, there was no statistical difference in  the average of the three experiments. Also, there are no explanations on why different result occurred from 3 trials. In general, three or more experiments are carried out in every in vitro study, taking into account the experimental environmental errors that may occur in the experiment, and whether the average value has statistical significance is explained. In addition, the authors also stated that qPCR and WB analysis were performed. However, authors did not explain enough about the results of WB. Even WB shows only one test result without mean or SD. The authors should perform more experiments including western blotting and show their opinion with statistical analysis. 

A:We did perform 3 independent biological replicates of our experiments that we called trials 1, 2 and 3. We clarified this point in the Methods section 4.3. The results were different from trial to trial, which is not unusual in cell culture experiments. For statistical analysis of the data we used a mixed model that allows for correlations between measurements using SAS statistical software, (section 4.7 of the Methods). This analysis showed a statistically significant difference in PAX6 expression (Table 1).

We did perform the western blot experiments in triplicate. The SDs are shown in the revised version of Figure 6.

Minor points

 Q:   -The abstract should be more specific and clearly rewritten. For example, you should equally comment on the results of qPCR and WB in order of experiment. It is also recommended that qPCR be described identically for each gene by gene / dose / time point sequence.

A:The length of the Abstract already exceeds the 200 words limit set by the journal; therefore, we tried to summarize only the major finding in the Abstract.

  Q:  - line 47-49. It is also necessary to make sure whether the "previous studies" have more than 2 references or not.

  A:                The sentence was corrected

  Q:  - line 99, In the results section, the period (.) must be deleted from "2.2 Effect of CT irradiation on expression markers."

A:We deleted the period

Reviewer 4 Report

Minor comments:

1. [Title]: Please change the title to a conclusive expression. Right now, it is hard to understand the effect is positive or negative from the title only.

2. [Abstract] Please add a conclusion sentence before future work.

3. [Abstract] Line 21-23 gene full names for PAX6, NES, and OCT4 should be provided.

4. [Introduction] Line 62 gene full names for PAX6, NES, and OCT4 should be provided.

5. [Result] Figure legend should have a brief title. Please add it for every figure.

6. Figure 6B. no statistic was provided. If it is only one experiment for Figure 6A, I suggest to remove it.

Author Response

Minor comments:

[Title]: Please change the title to a conclusive expression. Right now, it is hard to understand the effect is positive or negative from the title only.

We could not come up with a short phrase that adequately describes our paper’s findings. Therefore, we decided to keep the current title.

[Abstract] Please add a conclusion sentence before future work.

The length of the Abstract already exceeds the 200-word limit set by the journal; therefore, we summarized only the major finding in the Abstract.

[Abstract] Line 21-23 gene full names for PAX6NES, and OCT4should be provided.

The length of the Abstract already exceeds the 200-word limit set by the journal. We spelled out full names of the genes in the Introduction instead.

[Introduction] Line 62 gene full names for PAX6NES, and OCT4should be provided.

We included the full names of the genes in the revised Introduction.

[Result] Figure legend should have a brief title. Please add it for every figure.

We added short titles to the Figures.

Figure 6B. no statistic was provided. If it is only one experiment for Figure 6A, I suggest to remove it.

In the revised Figure 6 we presented standard deviations of 3 western blot experiments.

Round 2

Reviewer 3 Report

The author answered all issue from reviewers.